# mmWave Radar Sensors Fusion for Indoor Object Detection and Tracking

Xu Huang [1,2,*] , Joseph K. P. Tsoi [1] and Nitish Patel [1,*]

1 Department of Electrical and Computer Engineering, The University of Auckland, Auckland 1010, New Zealand; jgtsoi83@gmail.com
2 Faculty of Information Engineering, University of Shandong Ying Cai, Jinan 250104, China
* Correspondence: xhua559@aucklanduni.ac.nz or huangxu@ycxy.com (X.H.); nd.patel@auckland.ac.nz (N.P.)

**Abstract:** Indoor object detection and tracking using millimeter-wave (mmWave) radar sensors have received much attention recently due to the emergence of applications of energy assignment, privacy, health, and safety. Increasing the valid field of view of the system and accuracy through multi-sensors is critical to achieving an efficient tracking system. This paper uses two mmWave radar sensors for accurate object detection and tracking: two noise reduction stages to reduce noise and distinguish cluster groups. The presented data fusion method effectively estimates the transformation of the data alignment and synchronizes the result that can allow us to visualize the objects' information acquired by one radar on another one. An efficient density-based clustering algorithm to provide high clustering accuracy is presented. The Unscented Kalman Filter tracking algorithm with data association tracks multiple objects simultaneously in terms of accuracy and timing. Furthermore, an indoor object tracking system is developed based on our proposed method. Finally, the proposed method is validated by comparing it with our previous system and a commercial system. The experimental results demonstrate that the proposed method's advantage is of positive significance for handling the effect of occlusions at higher numbers of weak data and for detecting and tracking each object more accurately.

**Keywords:** millimeterwave; radar; detecting; object clustering; sensor fusion; tracking

## 1. Introduction

Precise multi-object detection and tracking are the predominant goals in any real-time location-based services (LBS), particularly indoor people activity tracking [1]. Indoor people activity tracking is a valuable solution to the problems that are associated with increasing populations, such as space availability, increased energy demands and health and safety issues, etc. [2]. Moreover, detection and tracking information can improve health and safety by allowing automated emergency systems and services to make more well-informed decisions. It can enhance the emergency services' response by providing them with real-time data on people's locations, how they are moving, and the densities of people at different places [3]. Thus, it will allow them to plan and execute rescue efforts more effectively, increasing the chances of survival for those at risk.

Ongoing research in object detection and tracking employs various sensing approaches and algorithms. Researchers typically use sensor technologies such as passive infrared sensors (PIR), ultra-wideband radar [4,5], LIDAR [6] and digital cameras [7,8]. However, all these technologies have challenges in terms of accuracy, privacy, and environmental robustness [9]. In this paper, we use a multi-sensors millimeter-wave (mmWave) radar sensor as the sensing technology [10,11] to build upon our previous work [12], which improved the tracking accuracy and the occluding problem (large group) for indoor people detection and tracking. The calibration and fusion are essential since sensor-specific data have different coordinates. We also introduce noise reduction, clustering, and advanced tracking algorithms between radar sensors and perform using data obtained from actual sensors.

This paper includes two main contributions. Firstly, we present a systematic approach for indoor object detection and tracking using multiple mmWave radar sensors. Two noise reduction stages reduce noise and distinguish cluster groups through density-based noise reduction and SNR data. The proposed data fusion method effectively estimates the homography that describes the transformation between two radar planes. We can visualize the objects' information acquired by one radar on another radar sequence using the data alignment and synchronized results. An efficient density-based clustering algorithm provides high accuracy. The Unscented Kalman Filter (UKF) tracking algorithm performs much better than the Extended Kalman Filter (EKF) in tracking accuracy and timing. Furthermore, an indoor object tracking system was developed based on our proposed method. By comparing the results to our previous work, the results show the method can handle the effect of occlusions at higher numbers of weak data and be more accurate. The paper is organized as follows. Section 2 details the research related to object detection and tracking, including the various types of sensor technologies and algorithms used. Section 3 presents the proposed architecture in detail, followed by the results evaluation and discussion in Section 4. Finally, the study is summarized and the potential future work is presented in Section 5.

## 2. Related Work

In recent years, practical implementations of machine learning and artificial intelligence (AI) methods using different sensing technology for object detection and tracking have begun to bear fruit outside of laboratory environments.

Passive Infrared Radiation sensors or PIR sensors use the change in the infrared heat radiation of bodies to detect objects. Such sensors were used in research conducted by the Auto-ID Laboratory in Japan, who reported an object measurement accuracy of 98.3% [13]. The disadvantages of such sensors are their narrow beam range and their limitation in detecting objects that are relatively stationary [14]. They are also limited due to the fact that PIR sensors require a wait time of at least 4 s before the state change is recognized; this produces inconvenience as it requires people to wait [15].

Ultrasonic sensors measure objects by examining the returned echo signals. Recent work in [4] proposed an object detector using multiple ultrasonic sensors. The system used the signal-to noise ratio (SNR) of a returned ultrasonic pulse as an indication of an object. A custom tracking algorithm was added onto the software pipeline to increase accuracy. The experimental accuracy results were around 80%. However, ultrasonic-based detection is plagued by significant safety concerns. Ultrasound waves affect people wearing hearing-aids and can be heard by a variety of animals, rendering ultrasonic based object detection unsuitable for indoor environments [2].

Due to the development of advanced embedded technology, onboard sensors like LiDAR and cameras have gradually become a standard configuration for object tracking. Researchers proposed many localization algorithms to obtain the accurate position of the object by matching the data from onboard sensors and the digital map. In [6], a localization method is proposed to estimate the pose of self-driving cars using a 3D-LiDAR sensor. With a map-matching method proposed to match the features to the map, a robust iterative closest point algorithm is utilized to deal with curb features, and a probability search method deals with intensity features. However, LiDAR is too expensive for home use, while throughout the development process, researchers have shown that haze can prove to be a big issue for LiDAR sensors [16,17]. In [7,8], the researchers used image data and depth learning methods to detect and track moving objects. These methods include decision trees, hidden Markov models, and convolutional neural networks such as YOLO and PoolNet. However, depth cameras only have a limited tracking range and accuracy while requiring a clear view and the right lighting conditions. Moreover, another critical problem with camera systems is their intrusive nature, leading to privacy concerns.

MmWave radar sensors are devices that use millimeter-wave signals, and it has been an exciting approach for object detection-related tasks due to its robustness and stability.

Its advantages include its protection of privacy, lack of dependence on light conditions, high accuracy (i.e., low false alarm rate), long detection range, and wide detection angle range [9]. INFINEON Technologies conducted a study using cardiopulmonary data gathered from mmWave sensors to determine occupancy [18]. The Doppler information extracted from the mmWave signals is then fed through band-pass filters to obtain the required cardiopulmonary data. The system's experimental accuracy was around 90% for between one and three people.

Moreover, the most related work is Texas Instruments' people counting and tracking system (TI) [19]. The system employs density-based clustering (DBSCAN) with an extended Kalman Filter (EKF). However, its accuracy is questionable due to its use of DBSCAN on a variable density point cloud. TI employed an EKF because their proposed system converted the polar radar measurement to Cartesian. The conversion was performed for ease of use but created an additional computational load that limited its embedded application. Additionally, our previous work aimed to improve TI's tracking system [12]. It improves the accuracy from TI's 96% for one person to 45% for five people to 98% for one person to 65% for five people. However, our previous system has limited accuracy when dealing with the occluding problem (a large group of people).

## 3. Materials and Methods

### 3.1. mmWave Radar Sensor Measurement System and Point-Cloud-Data Acquisition

3.1.1. mmWave Radar Sensor Measurement System

The mmWave radar sensor measurement system, including two millimeter-wave radar sensors (IWR1642BOOST) and a laptop control terminal, was established to acquire point-cloud data. The measurement system is shown in Figure 1. The main parameters are shown in Table 1.

The radar used in this paper, IWR1642BOOST [20], is an evaluation board containing a mmWave sensor with a Microcontroller Unit (MCU), which provides an end-to-end solution for object detection. IWR1642 is a single chip frequency modulated continuous wave (FMCW) radar from Texas Instruments (TI), which makes use of electromagnetic wave signals to determine the range, velocity, and angular information of matters. The fundamental concept of this type of radar is by first emitting a chirp signal ($Tx$) between 77 and 81 GHz, then capturing any signals reflected ($Rx$) by objects in its path, and mixing the $Tx$ and $Rx$ signals to produce an intermediate frequency ($IF$), taking a snapshot of the indoor location at a given point in time. The returned radar signal undergoes preliminary processing on the sensor, the output of which is a point cloud to tell the existence of an object.

The calculation process of the raw point-cloud data of the mmWave radar sensor measurement system for multiple objects includes the following steps [7]. By performing Fast Fourier Transform (FFT) to the signal $IF$, it can obtain the multiple peak frequency. Each peak with a different phase denotes the presence of an object at a specific distance (range) correspondingly. A second FFT performs the multiple steps to resolve things with a different speed (doppler). The third FFT on the small change of phases corresponding to the second FFT peaks to estimate the angle of arrival (AoA). By performing the above steps, the output is a point cloud.

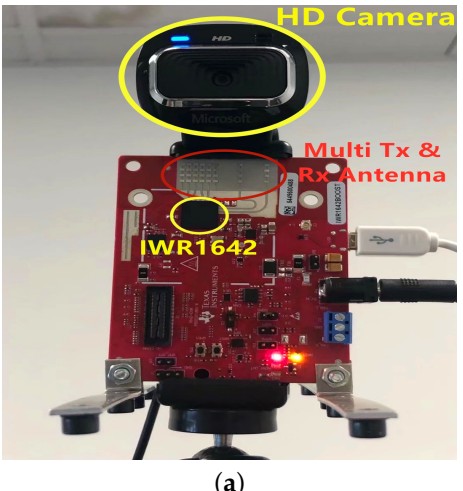
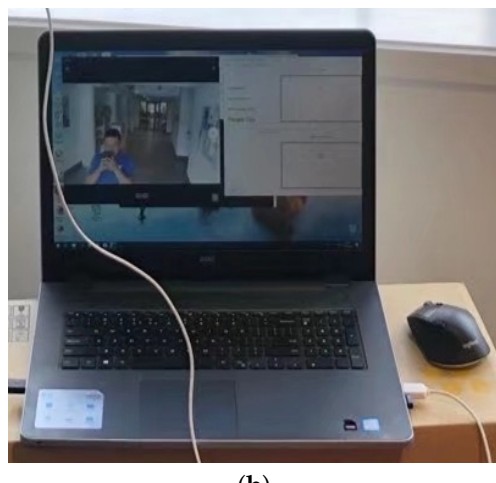

(**a**)                              (**b**)

**Figure 1.** Radar Measurement System. (**a**) mmWave IWR1642 BOOST Radar Sensor. (**b**) laptop Control Terminal.

**Table 1.** Main Parameters of IWR1642BOOST Radar Sensor.

| Performance Target | Parameter | Performance Target | Parameter |
| --- | --- | --- | --- |
| Number of RX | 4 | Range resolution | 4.9 cm |
| Number of TX | 2 | Maximun velocity | 18.64 Km/h |
| Field of view | 120° horizontal, 30° vert | Velocity resolution | 0.297 Km/h |
| Frequency | 77–81 GHz | Periodicity | 50 ms |
| Bandwidth | 4 GHz | Working voltage | 5 V |
| Range | 14 m | Power consumption | 2 V |

### 3.1.2. Point-Cloud Data Acquisition

The experiments were carried out at a research lab and a seminar room at the University of Auckland city campus. Experiments were conducted simulating various indoor activities to evaluate the algorithms' performance. Data were recorded simultaneously using two mmWave radar sensors mounted at a height of 1.8 to 2 m and a video camera to gather ground truth data. Each of the sensors has a field of view covers ranging from 1 to 6 m and azimuth from −60° to 60°. The room was selected to maximize the full range of the sensors. A 4.5 m by 6 m grid was drawn on the floor to contain the experiment within the sensors' range, which allowed us to control when occupants entered and left the site. The selected activities tested the sensor's capabilities and modeled real indoor scenarios. The tested activities included walking, standing, and walking cross, with each activity repeated multiple times, as shown in Figure 2.

The point cloud data were stored in a type of TLV (type-length-value) structure into a data frame. Hence, it was essential to parse the data to ensure reliable and accurate extraction in real-time. Each transmission's parsing began by reading the frame header into an array containing information such as the packet length, frame number, number of TLVs (number of data points in the point cloud), the header checksum etc. Then the TLV data, which contain the point cloud data, were read into another array. The TLV's size depended on the number of points detected in the field of view. The TLV header contained the TLV length, which was used to read the values by indexing the correct positions of the data frame. Figure 3 demonstrates a select frame of mmWave sensor raw point cloud data from a 3D view.

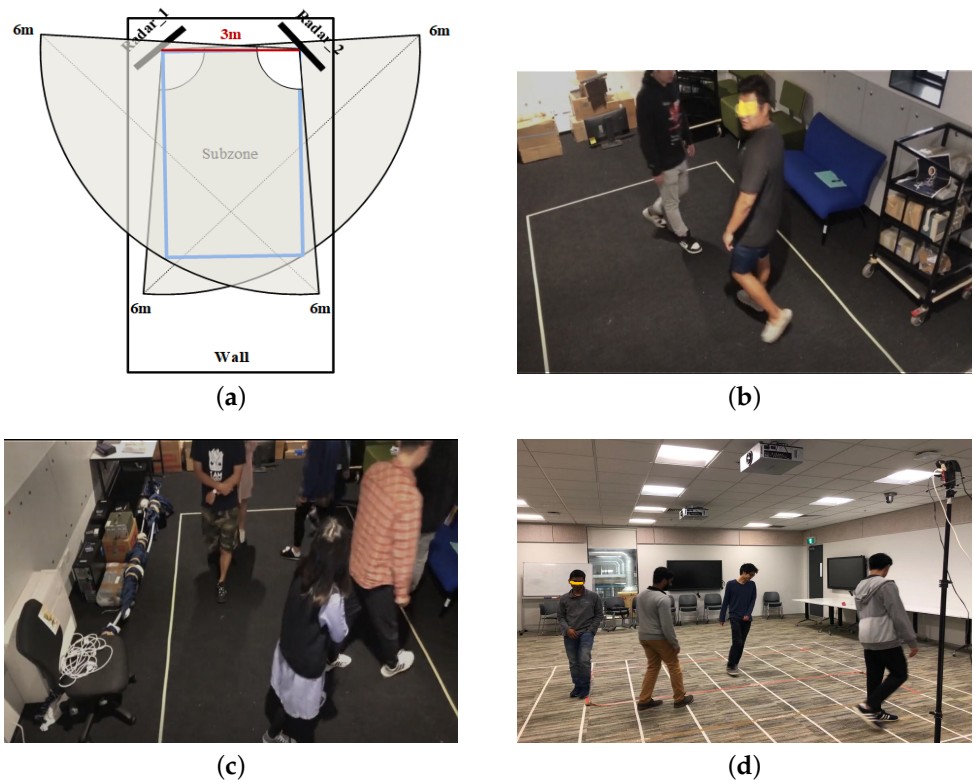

**Figure 2.** Point-cloud Data Collection. (**a**) Sensors Setup. (**b**) Research Lab: 2 People Walking Cross. (**c**) Research Lab:Large Group Walking. (**d**) Seminar Room.

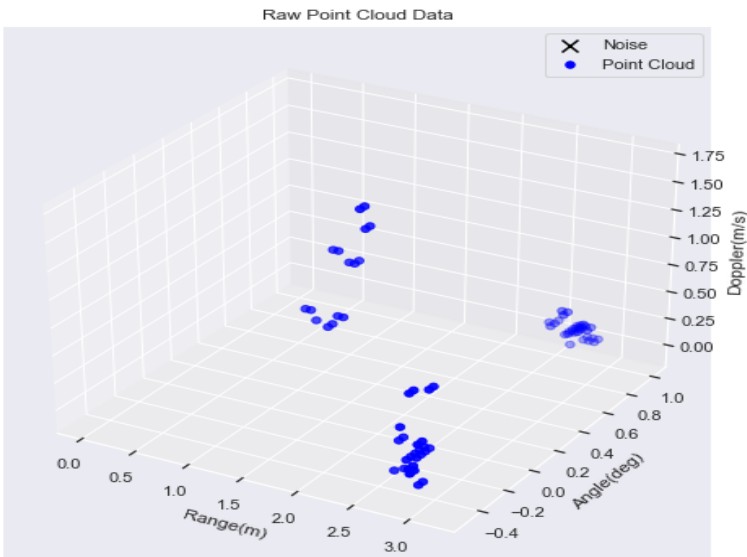

**Figure 3.** Raw Point Cloud Data.

### 3.2. System Design

Our system is a detecting and tracking system that processes and analyzes the unique properties of millimeter-wave radar. As Figure 4 shows, our research methodology includes four significant modules, including noise reduction, data fusion, clustering, and tracking. Firstly, the point cloud generated from the mmWave sensor is parsed and then processed for noise. Analyzing the point cloud generated then infers people's trajectories from a database.

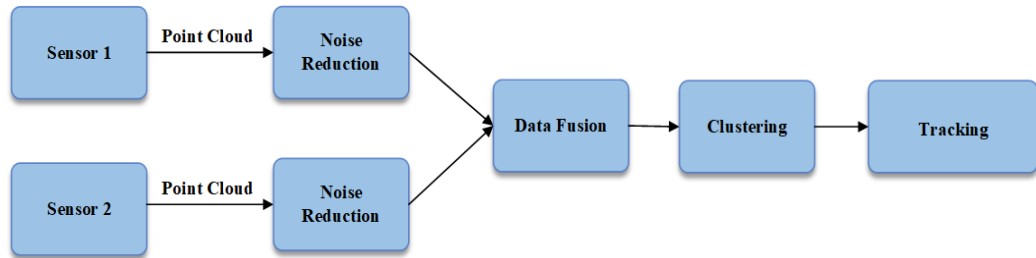

**Figure 4.** System Overview.

The operating pipeline is shown as follows:

1. Point Cloud Generation and Parsing. The FMCW Radar transmits millimeter waves to emit a radar signal and records the reflections in a scene at a given point in time. The returned signal undergoes preliminary processing on the sensor, then computes sparse points and removes those points corresponding to static objects, the output of which is a point cloud. The point cloud data are packaged into a data frame, and each frame has a header, followed by segments containing the point cloud information encoded in type-length-value (TLV) format;
2. Noise Reduction. The noise reduction stage is further divided into static clutter removal and signal-to-noise ratio (SNR) filtering;
3. Data Fusion. In this section, a data fusion method aims to combine measurements from multiple mmWave radar sensors that enable the tracker to efficiently utilize the radial measurements of objects from the radar module;
4. Clustering. In this section, a density-based clustering algorithm is designed to figure out how many objects are within the space at a given time;
5. Tracking. In this section, we associate the objects in consecutive frames and use multiple object tracking algorithms to maintain trajectories of different people.

### 3.3. Point Cloud Generation and Parsing

The point cloud data were packaged into a data frame using the TLV format. Hence, a parsing section must ensure reliable and accurate extraction before the data analysis process. Each data parsing began by reading the frame header into an array, containing information such as the packet length, frame number, number of TLVs (number of data points in the point cloud), etc. Then the TLV data, which contain the point cloud data, were read into another array. The TLV's size depended on the number of points detected in the field of view. The TLV header contained the TLV length, which was used to read the values by indexing the correct positions of the data frame.

### 3.4. Noise Reduction

#### 3.4.1. Static Clutter Removal

Static clutter removal was designed to eliminate as many of the static points as possible, that is, non-range changing (static) objects from the scene. The steps of the static clutter removal algorithm are listed as follows:

Step 1: Range processing performs Fast Fourier Transform (FFT) on Analog to Digital Converter (ADC) samples per antenna per chirp. FFT output is a set of range bins;

Step 2: Perform static clutter removal by subtracting the estimated Direct Current (DC) component from each range bin;

Step 3: Range processing results in local scratch buffers are Enhanced Direct Memory Access (EDMA) to the radar data cube with transpose.

#### 3.4.2. SNR Filtering

SNR filtering is the second stage of noise reduction. The higher the SNR of a point, the higher the certainty that the point corresponds to a person. The SNR filtering model aims to reduce the size of the remaining clusters to further improve performance and

create additional distance between objects. The impact of SNR filtering is explained and demonstrated further in Section 3.1.

In our SNR filtering method, the range of the sensor is divided into three regions. Observations in each region are filtered using a different SNR threshold. We presented a zone-based SNR filtering method to determine the optimal SNR ranges by using a multiplicative linear model (Equation (1)), as shown in Figure 5.

$$log(SNR_i) = \beta_0 + \beta_1 \times Range_i + \xi_i, \tag{1}$$

where $\beta_i$ is the regression slope. $\xi$ is the random unexplained error, and $\xi \sim iidN(0, \sigma^2)$.

We then used the lower bounds of its confidence intervals for the thresholds of the three different zones, including 1 m, 3 m, and 5 m. Table 2 contains the ranges and the associated SNR thresholds.

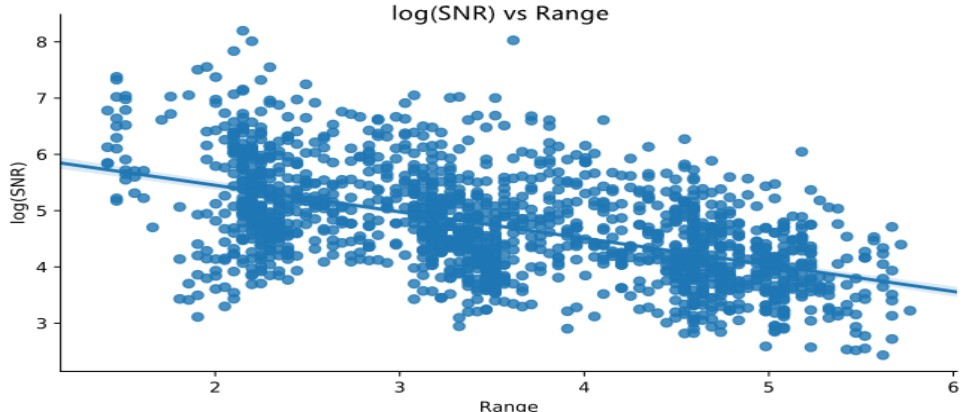

**Figure 5.** SNR vs. Range–Multiplicative Linear Model.

**Table 2.** Range corresponding with SNR thresholds.

| Range | SNR Threshold |
|-------|---------------|
| 1–3 m | 338 |
| 3–5 m | 139 |
| 5–6 m | 53 |

*3.5. Data Fusion*

The primary purpose of the data fusion technique is to combine measurements from multiple sensors that monitor common objects considering the uncertainty of individual sensors [21]. Ongoing research in data fusion technologies is mainly focused on vision-based sensors and millimeter-wave radar. Marco et al. [22] present a multi-modal sensor fusion scheme to estimate the three-dimensional vehicle velocity and attitude angles to enhance the estimation accuracy. Long et al. [23] and Guo et al. [24] separately use mmWave radar sensors and vision to detect surrounding obstacles and pedestrians. There are currently only limited studies on multiple mmWave radar data fusion.

Therefore, we propose a data fusion method based on the information from the mmWave sensors. We chose the information fusion method because it is simple in use and optimality. In what follows, we first explain the proposed calibration method to acquire the homography between the two radar sensors as described in Figure 6.

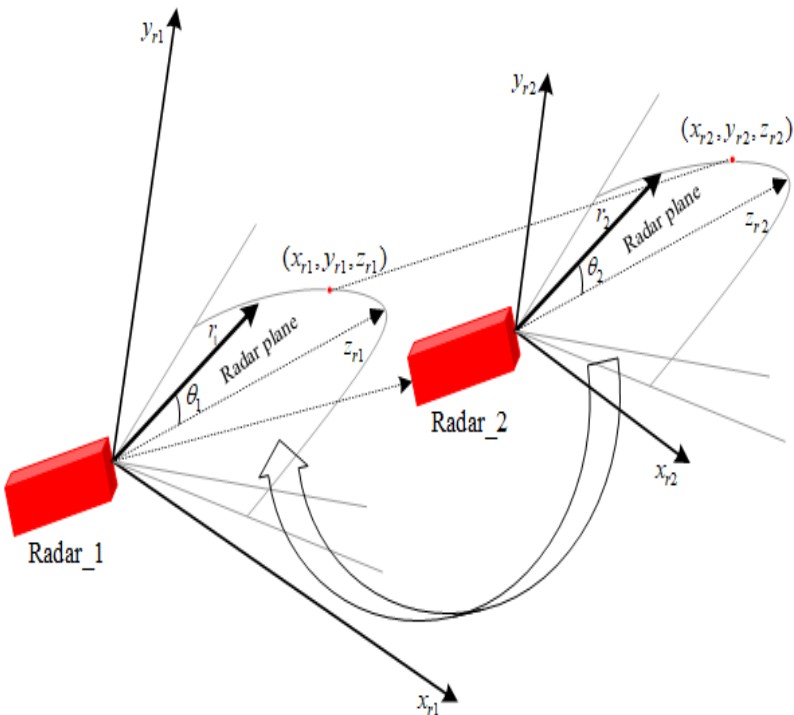

**Figure 6.** Sensor geometry.

### 3.5.1. Calibration

Denote the radar sensor coordinates by R1($x_{r1}$, $y_{r1}$, $z_{r1}$) and R2($x_{r2}$, $y_{r2}$, $z_{r2}$), respectively. If we fix the radar plane in $z_r = 0$, and set one of the radar sensors as the reference point (R1), then the relation between the two radar coordinates can be represented as:

$$\begin{bmatrix} x_{r2} \\ y_{r2} \\ 1 \end{bmatrix} = \begin{bmatrix} h_{11} & h_{12} & h_{13} \\ h_{21} & h_{22} & h_{23} \\ h_{31} & h_{32} & h_{33} \end{bmatrix} \begin{bmatrix} x_{r1} \\ y_{r1} \\ 1, \end{bmatrix} \tag{2}$$

where $H = [h_{i,j}]_{i,j=1,2,3}$ is the $3 \times 3$ homography which represents the coordinates relation between the two radar planes. Next, to figure out the homography matrix $H$, we must calculate the geometric transformations (translation and rotation) from Radar 2 to Radar 1.

(1) Translation. To translate the Radar 2 coordinate system to Radar 1 in the 2-D plane, we firstly set Radar 1 as the reference as described in Figure 7.

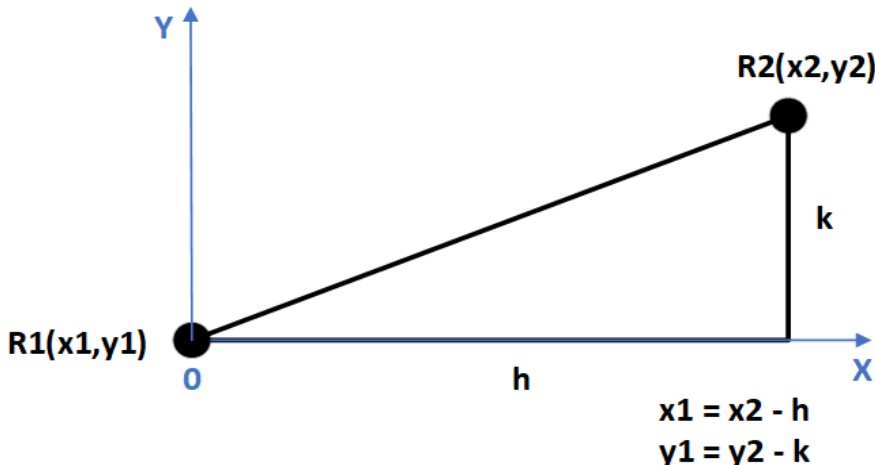

**Figure 7.** Translation.

The translation from R2 to R1 can be represented as:

$$\begin{bmatrix} x_{r2} \\ y_{r2} \\ 1 \end{bmatrix} = \begin{bmatrix} 1 & 0 & h \\ 0 & 1 & k \\ 0 & 0 & 1 \end{bmatrix} \begin{bmatrix} x_{r1} \\ y_{r1} \\ 1, \end{bmatrix} \tag{3}$$

where $x, y$ is the offset value in the X-Y plane between the two radar sensors.

(2) Rotation. Rotation is a relative quantity. The rotation of one point in the R1 plane is meaningful only concerning another point in R2. As such, rotation sensing capability requires two frames to make a measurement: measured and reference points. Again, we choose R1 for reference and the rotation relation described in Figure 8.

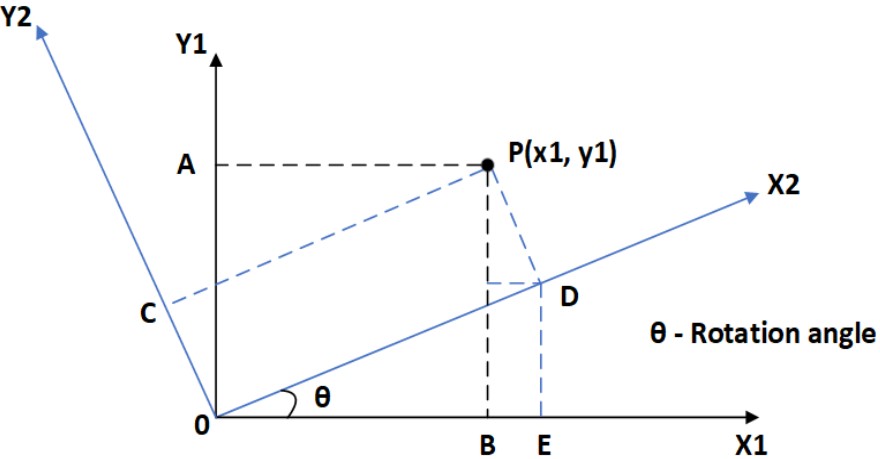

**Figure 8.** Rotation.

The rotation from R2 to R1 can be represented as:

$$\begin{bmatrix} x_{r2} \\ y_{r2} \\ 1 \end{bmatrix} = \begin{bmatrix} \cos\theta & \sin\theta & 0 \\ -\sin\theta & \cos\theta & 0 \\ 0 & 0 & 1 \end{bmatrix} \begin{bmatrix} x_{r1} \\ y_{r1} \\ 1, \end{bmatrix} \tag{4}$$

where $\theta$ is the rotation angle in X-Y plane between the two radar sensors.

(3) Combined transformations. The combined geometric transformations from R2 to R1 can be represented as:

$$\begin{bmatrix} x_{r2} \\ y_{r2} \\ 1 \end{bmatrix} = \begin{bmatrix} \cos\theta & \sin\theta & -h\cos\theta - k\sin\theta \\ -\sin\theta & \cos\theta & h\sin\theta - k\sin\theta \\ 0 & 0 & 1 \end{bmatrix} \begin{bmatrix} x_{r1} \\ y_{r1} \\ 1, \end{bmatrix} \tag{5}$$

where Equation (2) is represented as:

$$H = \begin{bmatrix} h_{11} & h_{12} & h_{13} \\ h_{21} & h_{22} & h_{23} \\ h_{31} & h_{32} & h_{33} \end{bmatrix} = \begin{bmatrix} \cos\theta & \sin\theta & -h\cos\theta - k\sin\theta \\ -\sin\theta & \cos\theta & h\sin\theta - k\sin\theta \\ 0 & 0 & 1. \end{bmatrix} \tag{6}$$

### 3.5.2. Data Alignment and Synchronize

Data fusion's primary purpose is to collect as much data as possible, giving more reliable performance. However, just collecting data from different data acquisition systems without efficient data alignment and a synchronised rule can seriously degrade the quality of an estimate even with a large amount of data [21].

However, even for two sensors configured to have identical sampling rates, minor differences from the actual sampling rate will occur on each sensor. They can result in gradual misalignment between samples from these two sensors. Sensors may also have

different internal timestamps at the start of recording due to clock drift, resulting in an initial time offset between recordings. Thus, we want to perform some alignment and synchronization on these recordings that account for both sampling rate drift and initial clock offsets. The synchronization needs to be both accurate and have a reasonable runtime.

Let us consider the two data acquisition systems, R1 and R2. Each of these sources provides a list of detection $r1 = \{a_1, a_2, \ldots, a_b\}$ and $r2 = \{b_1, b_2, \ldots, b_n\}$, respectively. To combine the information from these sources, we need to find the correlation between the detection in $r1$ and $r2$. All possible correlations can be expressed as a magnitude matrix $|r1 \times r2|$. In this paper, we use cross-correlation [25] to compare the two-time series and objectively determine how well they match each other and, in particular, when the best match occurs. The entire general pseudocode to the algorithm is shown in Algorithm 1.

---

**Algorithm 1:** Data Alignment and Synchronize with Perspective Transform.

---

**Require:**

    $\theta$—Sensor position phase from sensor 2

    $x_{r1}, y_{r1}$—Cartesian position dataset in sensor 2 perspective

    $x_{r2}, y_{r2}$—Cartesian position, horizontal and vertical dataset in sensor 1 perspective

    $h, k$—Horizontal and vertical position differences between sensor 1 and sensor 2

    $S$—Position magnitude of both sensors in sensor 1 perspective point of view

**Ensure:**

  1: Convert both datasets into Cartesian representation from sensor raw data.

  2: Project horizontal and vertical data set form Sensor 2 to Sensor 1 using perspective transform defined in (4). $x_{r2}, y_{r2} \leftarrow Perspective(x_{r1}, y_{r1}, \theta, h, k)$

  3: Evaluate objective position magnitudes from sensors 1 and 2 using the Pythagorean Theorem.
     $S \leftarrow Pythagorean(x, y)$

  4: Estimate sampling delay between both sensors, cross-correlated the prospectively projected sensor 2 data set with Sensor 1 data set.

  5: Obtain cross-correlation between sensor 1 and sensor 2.

  6: Get the time stamp of the maximum likelihood of the cross-correlated value.
     $SamplingDelay \leftarrow Expected(S_1, S_2)$

---

### 3.6. Clustering

A clustering method is the most efficient way to identify targets based on spatial information. Ti implemented DBSCAN at this stage and tried to identify the location information of targets [26], but failed to achieve high accuracy due to the varying density issue. During the literature review, a modified version of DBSCAN, VDBSCAN [27], was first tried to address this problem, yet there was no significant observation in accuracy improvement in this kind of system. Thus, we employed a density-based clustering algorithm to address this problem.

The beginning section of the density-based clustering algorithm works like the noise reduction module, as it treats each point as a node, and then calculates the distance matrix between itself and all the other nodes. However, in the clustering algorithm, if a node is within a distance threshold of 0.8 to the other nodes, then those nodes are extracted. If the number of points that have been associated together is greater than ten, it is then classified as a cluster. This process is iterated until all nodes are scanned and filtered. Ten was chosen as the minimum number of points through parameter optimization analysis, which is further discussed in Section 4.3.1.

The last stage of clustering was using k-means to convert the grouped clusters into individual centroids representing the objects. The k-means method calculates the mean of a given number of clusters, $k$. Through our previous stages, we know how many distinct groups of clusters there are. Hence, it uses this information to create k random cluster

points, which then it associates each centroid point with. The mean is then calculated, and the cluster points are shifted to the new means' positions. The process is repeated until no new changes in the associations are detected. The final means of the clusters represent the centroids of our objects.

The parameters associated with the clustering algorithm are:

- Minimum cluster size (minClusterSize): smallest number of points required to classify as a cluster;
- Minimum Maximum distance between points (maxDistance): largest distance possible between neighboring points to be associated within the same cluster.

As mentioned above, DBSCAN is not ideal because of the nature of the radar data. DBSCAN assumes a constant cluster density which is not the case with our data [28]. As shown in Figure 9, a person closer to the sensor (located at origin) is represented by a denser and more uniform cluster. In contrast, a further away person is represented by a less dense and more variable cluster.

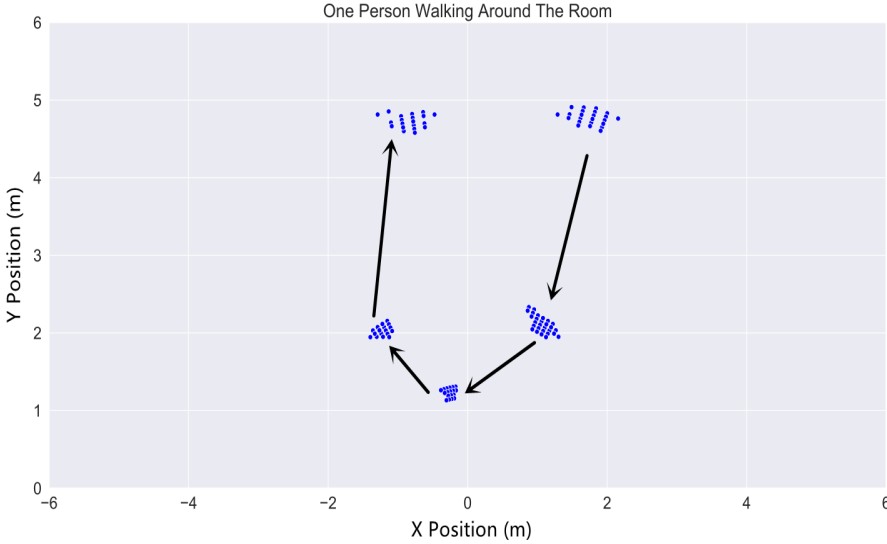

**Figure 9.** Comparison of density variations at different distance.

The density-based clustering algorithm we designed for this system can manage variable cluster densities. However, the limitation of this algorithm is it cannot handle noise as well as DBSCAN. Thus, creating the need for the prior noise reduction stage.

### 3.7. Tracking

The tracking stage is expected to input the point cloud data from the clustering layer, perform target localization, and report the target list to the user for visualization. Therefore, the tracker's output is a set of trackable objects with specific properties (like target ID, position, velocity, and other features) that the up layer can use.

### 3.7.1. Tracking Coordinate System

We transferred the radar data from the polar system to the Cartesian system for convenience during the data fusion stage. In the tracking stage, we chose to track in Cartesian coordinates to keep the same page with data fusion and for target motion extrapolation. However, we still decided to keep measurement inputs in polar coordinates to avoid error coupling. We used Unscented Kalman Filter (UKF) to linearize the dependencies between tracking states and measurement vectors.

The tracking in Cartesian coordinates is illustrated in Figure 10 [19]. There is a single reflection point at time *n*. Multiple reflection points represent real-life radar objects. Each

point is represented by range $r$ ($R_{min} < r < R_{max}$), angle $\theta$ ($-\theta_{min} < \theta < \theta_{max}$), and radial velocity $\dot{r}$ (range rate).

The angular location coordinates are converted to Cartesian coordinates using equations as below:

$$x = r\cos(\pi/2 - (\alpha + \theta)) = r\sin(\alpha + \theta) \tag{7}$$

$$y = r\sin(\pi/2 - (\alpha + \theta)) = r\cos(\alpha + \theta). \tag{8}$$

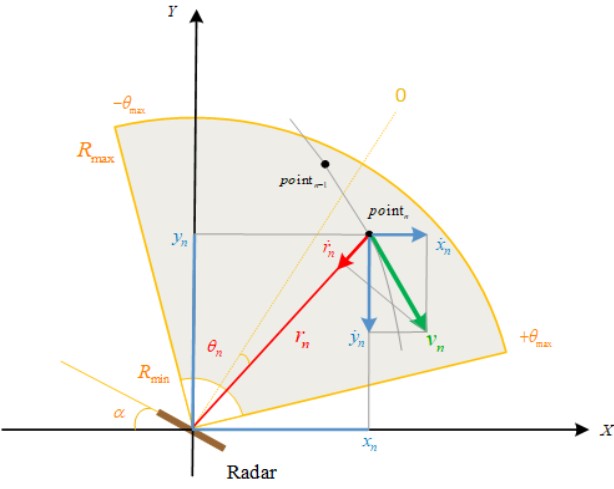

**Figure 10.** Tracking in Cartesian coordinates.

### 3.7.2. Unscented Kalman Filter + Constant Velocity Model

We opted for the UKF, as it has not been researched in the domain of mmWave tracking, and it can improve accuracy by avoiding the EKF's process errors caused by linearization by taking only one point, i.e., mean, and approximate transfer of the coordinates from Polar to Cartesian. The UKF is well-suited for tracking indoor people using a constant velocity (CV) model, and we also considered an acceleration model by random noise. The UKF process is as follows:

The state of the Kalman filter at time instant $n$ is defined as:

$$X(n) = FX(n-1) + Q(n), \tag{9}$$

where the state vector $X(n)$ is defined in Cartesian coordinates,

$$X(n) = [x(n)\ y(n)\ \dot{x}(n)\ \dot{y}(n)]^{T}. \tag{10}$$

$F$ is a transition matrix,

$$F = \begin{bmatrix} 1 & 0 & \Delta t & 0 \\ 0 & 1 & 0 & \Delta t \\ 0 & 0 & 1 & 0 \\ 0 & 0 & 0 & 1, \end{bmatrix} \tag{11}$$

where $\Delta t$ is the mmWave sensor sampling time interval and was set to 50 ms. $Q$ is the system process noise covariance matrix.

The input measurement vector $z(n)$ includes range, angle and radial velocity,

$$z(n) = [r(n)\ \theta(n)\ \dot{\theta}(n)]^{T}. \tag{12}$$

The relationship between the state of the Kalman filter and the measurement vector is expressed as:

$$z(n) = H(X(n-1)) + R(n), \tag{13}$$

where R(n) is the measurement noise covariance matrix. Since our indoor tracking system only provides position not velocity, the measurement function is:

$$H = \begin{bmatrix} 1 & 0 & 0 & 0 \\ 0 & 1 & 0 & 0. \end{bmatrix} \tag{14}$$

Predict Step

(1) Calculate Sigma Points. The number of sigma points depends on the dimensional of the system $N$. The general formula is $2N + 1$, where $N$ denotes the dimensions.

$$\begin{aligned} \chi_0 &= x \\ \chi_i &= x + \left(\sqrt{(N+\lambda)P}\right)_i \quad for \quad i = 1, ..., N \\ \chi_i &= x - \left(\sqrt{(N+\lambda)P}\right)_{i-n} \quad for \quad i = N+1, ..., 2N, \end{aligned} \tag{15}$$

where Calligraphic $\chi$ denotes the Sigma point matrix, $x$ is the mean of the Gaussian and $\lambda$ is the spreading parameter that tells how far from the mean we should choose our sigma points. Here, we define $\lambda = \alpha^2(N + \kappa) - N$. $P$ is a covariance matrix. The $i$ subscript chooses the $i^{th}$ column vector of the matrix. In other words, we scale the covariance matrix by a constant, take its square root, and ensure symmetry by adding and subtracting it from the mean. One of the sigma points is the mean, and the rest we calculate based on the above equations.

(2) Computing Weights of Sigma Points. We use one set of weights for the means, and another set for the covariance. The weights for the mean of $\chi_0$ is computed as:

$$w_0^m = \frac{\lambda}{N+\lambda}. \tag{16}$$

The weight for the covariance of $\chi_0$ is:

$$w_0^c = \frac{\lambda}{N+\lambda} + 1 - \alpha^2 + \beta. \tag{17}$$

The weights for the rest of the sigma points $\chi_0 \ldots \chi_{2N}$ are the same for the mean and covariance. They are:

$$w_i^m = w_i^c = \frac{1}{2(N+\lambda)} \quad for \quad i = 1, \ldots, 2N. \tag{18}$$

(3) Choices for the Parameters. We set $\beta = 2$ for Gaussian problem choice, $\kappa = 3 - N$, and $0 \le \alpha \le 1$ is an appropriate choice for $\alpha$, where a larger value for $\alpha$ spreads the sigma points further from mean.

(4) Unscented Transforming Sigma Points and Calculate new Mean and Covariance.

$$\begin{aligned} x' &= \sum_{i=0}^{2N} w_i^m g(\chi_i) \\ P' &= \sum_{i=0}^{2N} w_i^c (g(\chi_i) - x')(g(\chi_i) - x')^T + Q, \end{aligned} \tag{19}$$

where $x'$ is predicted mean, $P'$ is predicted covariance, $w$ is weights of sigma points, $g$ is process model (non linear function), and $Q$ is the system process noise matrices.

Update Step

In the update step, Kalman filters perform the update in measurement space. Then we have a measurement coming from the sensor. To compute the difference between our predicted values of mean and covariance and actual values of mean and covariance, we

convert the sigma points of the prior into measurements using a measurement function $Z = h(\chi)$. The equations below show the calculated mean and covariance of these points using the unscented transform.

$$
\begin{aligned}
x_z &= \sum_{i=0}^{2N} w_i^m Z_i \\
P_z &= \sum_{i=0}^{2N} w_i^c (Z_i - x_z)(Z_i - x_z)^T + R,
\end{aligned}
\tag{20}
$$

where $Z$ has transformed sigma points in measurement space, $h$ is a function that maps our sigma points to measurement space, which can transform our state-space to measurement space to equate them in the same units. $x_z$ is the mean in the measurement space, $P_z$ is covariance in the measurement space, and $R$ is the measurement noise matrix.

Next, we compute the residual and Kalman gain. The residual of the measurement $z$ is:

$$
y = z - x_z.
\tag{21}
$$

To compute the Kalman gain, we first calculate the cross-covariance of the state and the measurements, which is defined as:

$$
P_{xz} = \sum_{i=0}^{2N} w_i^c (\chi_i - x')(Z_i - x_z)^T.
\tag{22}
$$

Then the Kalman gain $K$ is defined as:

$$
K = P_{xz} P_z^- 1.
\tag{23}
$$

Finally, the new state estimate using the residual and Kalman gain and the new covariance are computed as:

$$
\begin{aligned}
x &= x' + Ky \\
P &= P' + K P_z K^T.
\end{aligned}
\tag{24}
$$

### 3.7.3. Data Association

The Kalman filter can only track a single person at a time. Therefore, a data association method to match people between discrete frames of information is required to enable multiple target tracking, as shown in Figure 11. Data association involved calculating the distance between each object's previous frame location and all the new frames' measured locations. Hence, each object in the previous frame will be associated with all the new objects' data and their distances. After that, the global nearest neighbor is found, i.e., the object with the smallest distance. The object is then associated with the new measured data and deleted from the distance matrix. This process is iterated until all the previous frame's objects have been associated. All unassociated objects are classified as new entrants.

After data association is performed, the associated centroids can be passed through the update step of the Unscented Kalman filter, producing location estimates of each person.

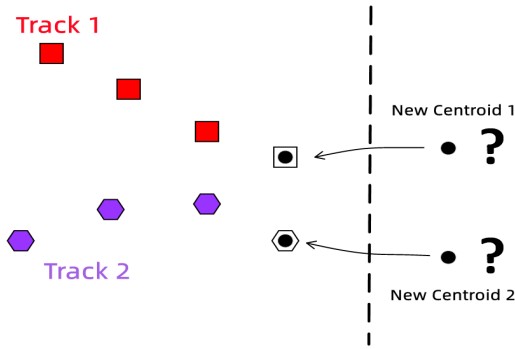

**Figure 11.** Data Association Visualized.

## 4. Evaluation

### 4.1. SNR Range Optimization Analysis

As explained in Section 3.4.2, noise reduction is performed using SNR values. We performed a statistical analysis to determine the optimal SNR ranges by fitting a multiplicative linear model, using Equation (1). We then used the lower bounds of its confidence intervals for the thresholds of the three different zones, including 1 m, 3 m, and 5 m. Hence, this resulted in the SNR thresholds of 338, 139, and 53 for 1–3 m, 3–5 m, and 5–6 m, respectively. Using different SNR thresholds is required as, on average, the SNR of observed people increases with range. The demonstration of SNR filtering is shown in Figure 12. The two targets were separated by the SNR filtering process when they were crossing each other.

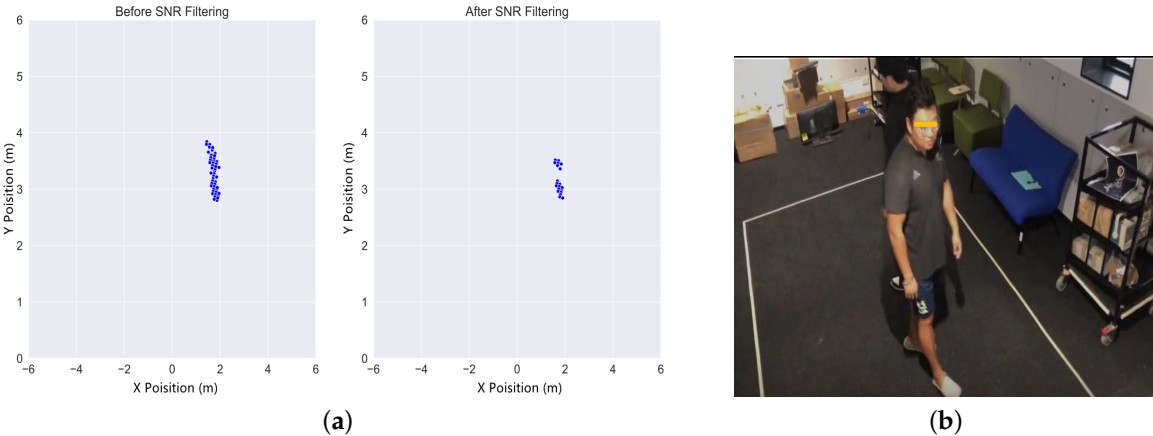

**Figure 12.** Demonstration of SNR Filtering. (**a**) Before and After SNR Filtering Process. (**b**) Experiment Site.

### 4.2. Data Fusion Evaluation

A data fusion method was used to simultaneously process the data of the two radar sensors used for indoor people detecting and tracking. The radar sensors' data obtained from the actual experiment matched in the manner described in Section 3.1.

#### 4.2.1. Calibration

Figure 13 shows the raw data from the two radar sensors for one person walking around ("OnePersonWalking.mat") obtained through the experiment (Section 3.1.2).

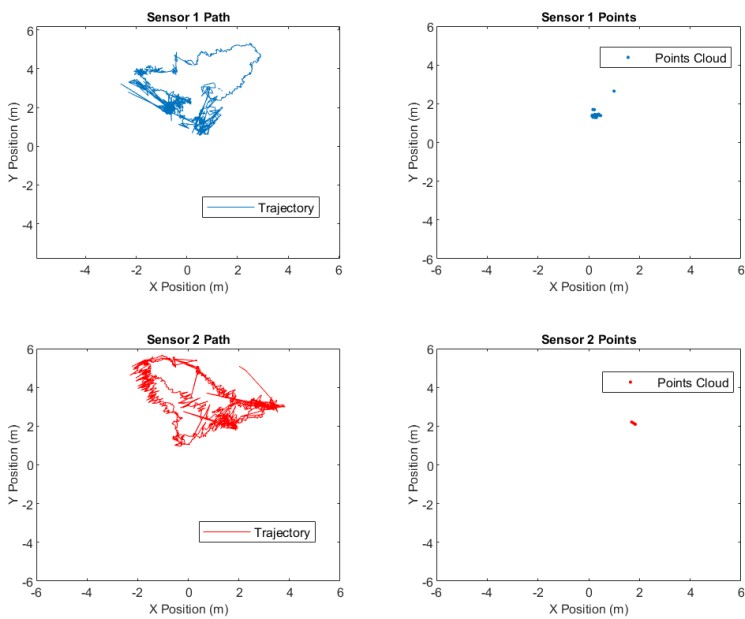

**Figure 13.** Raw Data Plotting Before Calibration.

The point cloud data obtained through the experiment do not coincide with the coordinates of each sensor. For the coordinate matching, the transformation matrix $H$ (the coordinates relation between the two radar planes) obtained using the experimental data is as follows. Where $h = -3$ (m) is horizontal shift, $k = 3$ (m) is vertical shift but upside-down angle, $\theta = -90/180 \times \pi$ is rotation angle offset.

$$H = \begin{bmatrix} \cos\theta & \sin\theta & -h\cos\theta - k\sin\theta \\ -\sin\theta & \cos\theta & h\sin\theta - k\sin\theta \\ 0 & 0 & 1 \end{bmatrix} = \begin{bmatrix} -3.6732 \times 10^{-6} & -1 & 3 \\ 1 & -3.6732 \times 10^{-6} & 6 \\ 0 & 0 & 1. \end{bmatrix} \tag{25}$$

In this experiment, we used 7104 frames data (3717 frames from Sensor 1, 3387 frames from Sensor 2) to obtain the calibration result. As a result, we transformed the object coordinate of Sensor 2 into Sensor 1 coordinates by using Equation (5). Figure 14 shows the combined transformation of the two sensors.

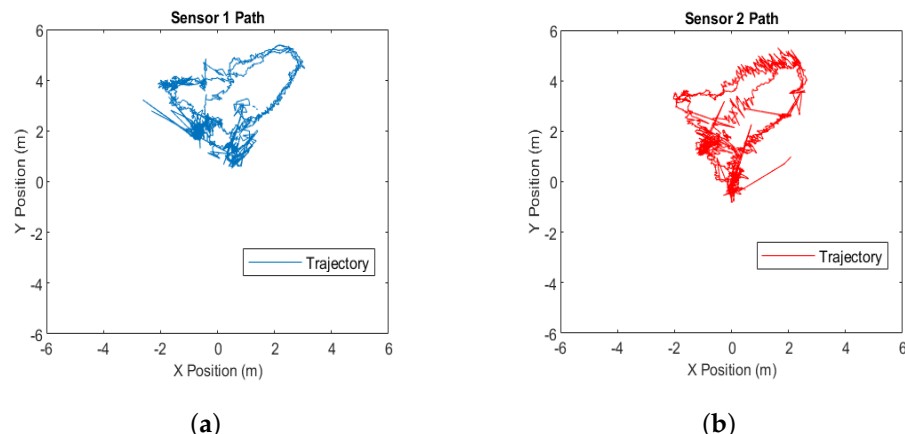

(**a**)            (**b**)

**Figure 14.** Combined Transformation After Calibration. (**a**) Sensor 1 Trajectory. (**b**) Sensor 2 Trajectory.

4.2.2. Synchronize Result

As mentioned in Section 3.5.2, we proposed a data alignment and synchronized algorithm to obtain cross-correlation and the delay frames between the two sources. The

calculated delay frame number and the final data fusion result are shown in Figure 15. As can be seen, the sensor 2 delayed frame number is 468 frames, and the sensor 2 delayed time is $T_d = \frac{468\,frames}{20\,frames/s} = 23.4$ s. Moreover, the transformed trajectory of the two sensors is not the same, i.e., using two sensors for indoor people tracking can obtain much more point cloud data than a single one.

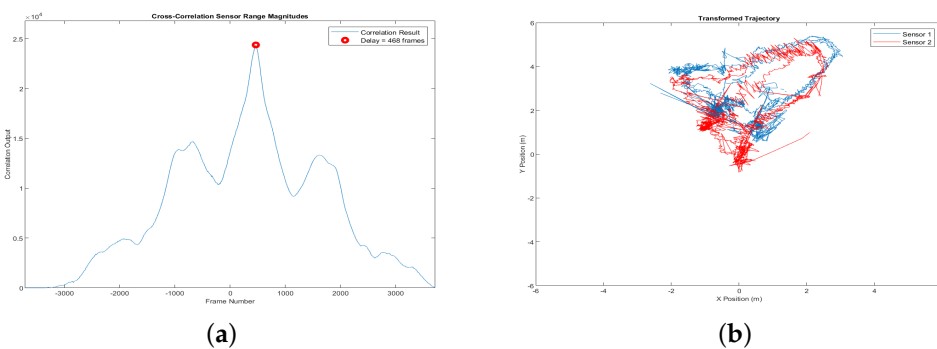

(**a**)          (**b**)

**Figure 15.** Synchronize Result. (**a**) Correlation and Find the Delay Frames. (**b**) Transformed Trajectory of the Sensors.

*4.3. Clustering Evaluation*

4.3.1. Clustering Parameter Optimization

The parameters minClusterSize and maxDistance were optimized by running various combinations of the parameters through the dataset collected. Pairs of maxDistance and minClusterSize versus accuracy are shown in Figure 16. The highlighted point in Figure maximized accuracy with an accuracy value of 96.8% and minClusterSize of 10 points and maxDistance of 0.8 m.

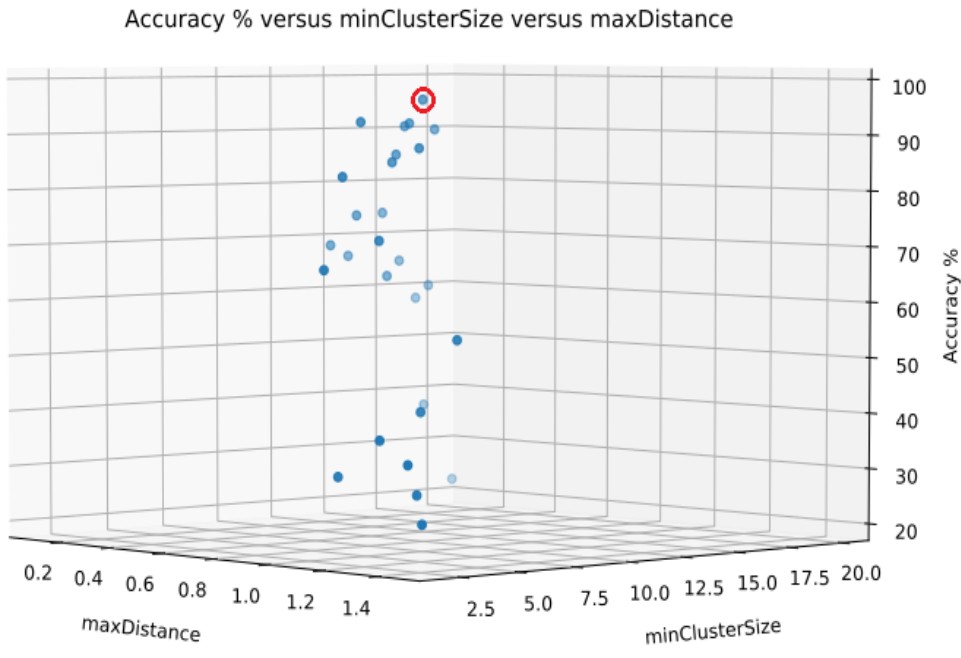

**Figure 16.** View of 3D scatterplots Showing Parameters That Maximized Accuracy.

4.3.2. Clustering Accuracy

We compared our current system with our previous work [12] to evaluate the clustering accuracy. The accuracy was calculated by comparing the number of correctly classified samples to the total number of samples collected. Figure 17 shows that with between

one and five people, the dual sensors system's accuracy ranges between 84% and 99%. The previous system has an accuracy of between 65% and 98%. However, it also shows that the performance of both systems degrades as the number of people increases. As the number of people increases, more objects begin impeding one another, decreasing accuracy. Moreover, we noticed that the amount of missing/corrupted data sent from the millimeter-wave sensor increases with a higher number of people. Missing/corrupted frames are a significant reason for the discrepancy in accuracy between both systems.

### 4.4. Tracking Evaluation

### 4.4.1. Tracking Algorithms

To evaluate the tracking algorithms based on the data association algorithm and Unscented Kalman Filter (UKF), we mainly focus on the tracking accuracy and timing against Kalman Filter (KF) and Extended Kalman Filter (EKF) and and use a public dataset [29] with ground truth. This is a famous public dataset for Kalman Filter fusing radar sensor measurements. We ran through various options for the UKF, EKF, and KF weighting matrices initialization and optimization and chose the best performing combinations for comparison.

By contrast, UKF, EKF, and KF are employed to track the same target. Figure 18 shows the filter results between UKF, EKF, and KF using a public dataset, and Table 3 shows the comparison of Root Mean Squared Error (RMSE) and timing between the UKF, EKF, and KF with a public dataset.

As can be seen, both KF, EKF, and UKF can estimate unmeasurable system states and smooth out the process/measurement noise very well. However, in terms of algorithmic accuracy, UKF performs much better than the KF and EKF since UKF linearizes a nonlinear function around multiple points.

**Table 3.** Comparison Between the UKF, EKF, and KF Using Public Datasets.

| Algorithms | Total Frames | Object Number | RMSE | Timing (ms) |
| :---: | :---: | :---: | :---: | :---: |
| KF | 1224 | 1 | 0.1025 | 171.9 |
| EKF | 1224 | 1 | 0.0822 | 265 |
| UKF | 1224 | 1 | 0.0365 | 359.4 |

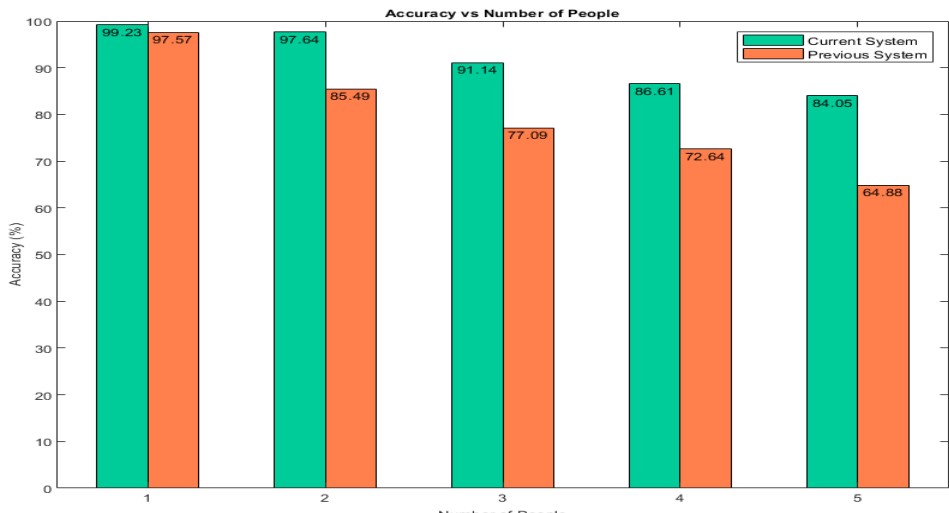

**Figure 17.** Accuracy vs Number of People.

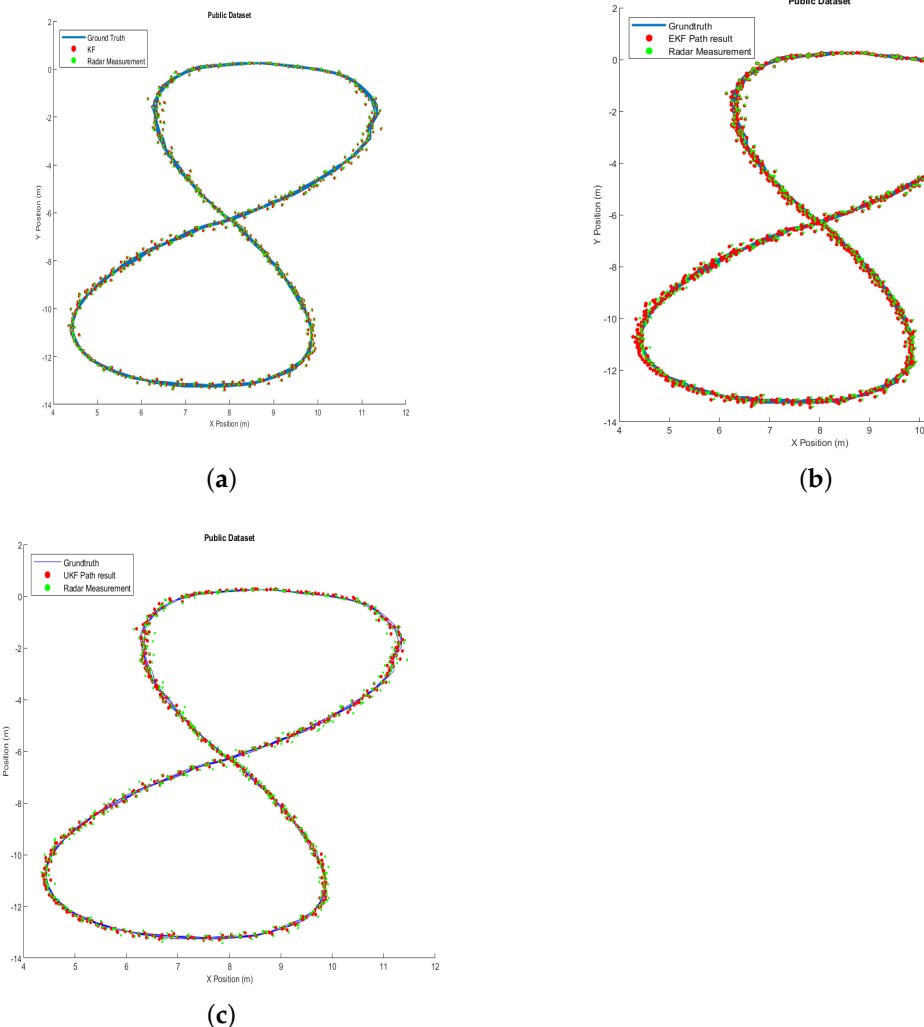

**Figure 18.** UKF vs. EKF vs. KF. (**a**) KF. (**b**) EKF. (**c**) UKF.

#### 4.4.2. Tracking Accuracy

Two datasets were collected from a walking TurtleBot2 robot [30] (model of a lower object) at the three position-known locations (sensor 1 coordinate system as a reference) to evaluate the tracking accuracy. The moving speed of TurtleBot2 was configured as 0.35 m/h for linear and 3.1 m/h for angular. The experiment site, sensors setup, and sensors tracking shot after fusion are shown in Figure 19. A 4 m by 4 m grid was drawn on the floor to contain the experiment within the sensors' range, which allowed us to control when the robot reached and moved at the three position-known locations.

Then, we ran those datasets through our current dual sensors system, the previous work, and a commercial system from Texas Instruments (TI) and calculated the Root Mean Square Error (RMSE) in X Y directions. The location coordinates from sensor 1 are shown in Table 4. Table 5 shows that the average position errors of our current system were 0.2136 m in the x-direction and 0.2290 in the y-direction. In comparison, the average position errors of the previous system were 0.5401 m in the x-direction and 0.5601 m in the y-direction. The average position errors of TI's system were 0.5481 m in the x-direction and 0.5903 m in the y-direction.

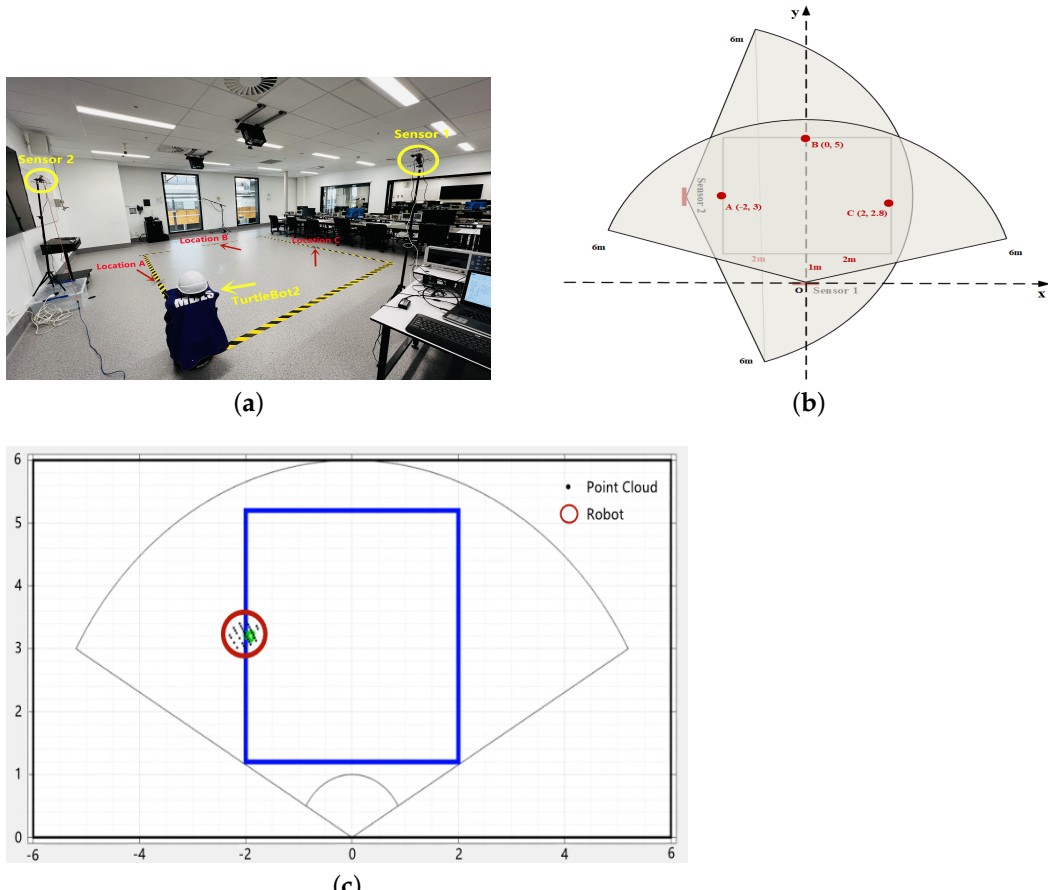

**Figure 19.** Tracking Accuracy Experiment. (**a**) Experiment Site. (**b**) Sensors Setup. (**c**) Tracking Shot.

During the experiment, occlusions and weak data occured when the robot entered a range in which the sensor returned weak signals or was lost from the view of the sensor by a height lower than a human being. This resulted in the robot not appearing in the point cloud due to the lower amount of signal reflections. Although the use of millimeter waves and advanced tracking algorithm theoretically increased the detectability of the smallest of movements, it would still only appear with little to no presence in the point cloud as there was still a smaller number of reflections in total. Data fusion from two millimeter-wave sensors is a good solution that decreases the effect of occlusions at higher numbers of weak data and increases the effective field of view. The dual sensors' fusion system can improve the tracking accuracy dramatically.

**Table 4.** Marked Ground Truth for Tracking Accuracy Comparison.

| Location | X (m) | Y (m) |
|:---:|:---:|:---:|
| A | −2 | 3 |
| B | 0 | 5 |
| C | 2 | 2.8 |

**Table 5.** RMSE Comparison Between Two Systems.

| DataSet | Total Frames | Systems | RMSE X (m) | RMSE Y (m) |
|---------|--------------|---------|------------|------------|
| Location A | 2826 | TI System | 0.6630 | 0.6983 |
| Location A | 2657 | Previous System | 0.6505 | 0.6818 |
| Location A | 5236 | Current System | 0.2530 | 0.2648 |
| Location B | 2863 | TI System | 0.4682 | 0.4830 |
| Location B | 2963 | Previous System | 0.4316 | 0.4641 |
| Location B | 5089 | Current System | 0.1553 | 0.1892 |
| Location C | 1626 | TI System | 0.5331 | 0.5895 |
| Location C | 1864 | Previous System | 0.5382 | 0.5343 |
| Location C | 3395 | Current System | 0.2326 | 0.2331 |

## 5. Conclusions

In this paper, a mmWave radar sensors fusion system for indoor object detecting and tracking is designed based on the proposed data process algorithms. Our methodology is processed in the order of point cloud generation and parsing, noise reduction, data fusion to combine measurements from multiple sensors, clustering into clusters, and referencing to identify the centroids, then tracking the centroids by using Unscented Kalman Filter (UKF). The experiments were set up at different data collection sites modeling various indoor scenarios. Compared to our previous system and a commercial system, this fusion system can handle the effect of occlusions at higher numbers of weak data and detect and track each object more accurately.

The real-time solution of our system is constrained by processing time, which will only improve as processing power advances. The accuracy and strength of the data decreased within a few meters of the range; with further advancements in mmWave technology, this can be improved. The main challenge with clustering is recognizing noise from objects. Hence, a deep learning approach could be an exciting avenue for future research to improve accuracy and classify various species' objects.

**Author Contributions:** Conceptualization, X.H. and N.P.; methodology, X.H.; software, J.K.P.T. and X.H.; validation, X.H. and J.K.P.T.; resources, X.H. and N.P.; writing—original draft preparation, X.H.; writing—review and editing, X.H. and N.P.; visualization, X.H. and J.K.P.T.; project administration, N.P.; funding acquisition, X.H. and N.P. All authors have read and agreed to the published version of the manuscript.

**Funding:** This research was funded by National Statistical Science Research Project, China, Grant number 2019LY82.

**Acknowledgments:** The authors wish to acknowledge the technical support from Hasnain Cheena and Abin Thomas.

**Conflicts of Interest:** The authors declare no conflict of interest.

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
