# Peer review of "mmWave Radar Sensors Fusion for Indoor Object Detection and Tracking"

_electronics, doi:10.3390/electronics11142209_

Round 1

Reviewer 1 Report

In this paper, the authors present a strategy to fuse two mmWave radar sensors for accurate object detection and tracking, and an indoor object tracking is completed by the UKF. Real datasat is provided to verify the advantage of the proposed algorithm. Overall, the topic of this paper is interesting. I have the following concerns:

1. Abstract: It is a bit long and it is better 180~200 words. The description of the method can be condensed.

2. Introduction: P2, line 33: many signal processing method can also be introduced, e.g., [1-2]. Please add these references in the ‘ultra-wideband radar’:

[1] X. Hua, Y. Ono, L. Peng and Y. Xu, "Unsupervised Learning Discriminative MIG Detectors in Nonhomogeneous Clutter," in IEEE Transactions on Communications, 2022, doi: 10.1109/ TCOMM.2022.3170988.

[2] H. Oh and H. Nam, "Energy Detection Scheme in the Presence of Burst Signals," in IEEE Signal Processing Letters, vol. 26, no. 4, pp. 582-586, April 2019.

3. Introduction: The outline of this paper should be added.

4. It’s better to give a clear comparison about various filter methods, such as UKF, KF, EKF.

5. Conclusions: The authors should point out the potential limitations of their work.

Reviewer 2 Report

1. The problem definition, motivation and more recent related work must be added in the introduction section. My suggestion is to divide the introduction into three subsections: motivation and incitement, literature review and contribution and paper organization.

2. There are a lot of citations but they are grouped without detailed information how their contents affect the plot of the presented research. It should be divided and separately described.

3. It helps to appreciate the paper by having a related work section. Please include recent research from 2020 and 2021 if possible. The manuscript should list the drawbacks to the present work and research scope in this area.

4. There is no discussion of user requirements, technological options and support for the decisions made at the design. The authors should include more technical details and explanations.

5. The proposed processes should be revised in a more formal pseudocode template. The authors should include more technical details and explanations.

6. The authors need to interpret the meanings of the variables. Some parameters and their values are unknown. It would be better to show all these parameters and explain the reason for those numbers in the table.

7. The comparison to other improved schemes (more current literature in the area) is required. This paper should summarize those results and give a comprehensive performance comparison with previous works.

8. It needs to highlight the research main contribution with some brief indications and numerical improvement percentages to keep with the reader.

9. The conclusion and future work part can be extended to have a better understanding of the approach and issues related to that which can be taken into consideration for future work.

10. The authors did not provide solid achievements in this manuscript since this paper seems to be a somewhat incremental piece of work based on earlier research results [A].

[A] X. Huang, H. Cheena, A. Thomas and J. Tsoi, Indoor Detection and Tracking of People Using mmWave Sensor, Journal of Sensors, vol. 2021, pp. 1-14, 2021.

Round 2

Reviewer 1 Report

The authors have addressed all my concerns.

Reviewer 2 Report

This paper has edited and revised according to the reviewer's suggestions.